# Predictors of Early Mandarin Chinese Character Reading Development

**DOI:** 10.3390/children9121946

**Published:** 2022-12-12

**Authors:** Han Yuan, Eliane Segers, Ludo Verhoeven

**Affiliations:** 1Faculty of Education, Shenzhen University, Shenzhen 518061, China; 2Behavioural Science Institute, Radboud University, Montessorilaan 3, 6525 HR Nijmegen, The Netherlands

**Keywords:** visual perception skills, phonological awareness, longitudinal, Chinese character reading

## Abstract

In the present study, we investigated the development of Chinese character reading and its predictors in 55 children from K3 (the last year of kindergarten) to G1 (first grade) in Mainland China. It was examined to what extent first graders’ Mandarin Chinese character reading was related to their phonological awareness, Pinyin letter knowledge, and visual perception skills in kindergarten. The results showed that phonological awareness, Pinyin letter knowledge, and Chinese character reading improved from kindergarten to Grade 1, with Pinyin showing ceiling effects in Grade 1. Children’s character reading in first grade was not predicted from phonological awareness in kindergarten. However, visual discrimination showed an indirect effect on Grade 1 character reading via its effect on kindergarten character reading. It can be concluded that both kindergarten visual discrimination and character reading ability facilitate first-grade reading ability for children in Mainland China.

## 1. Introduction

Numerous studies have focused on the metalinguistic and cognitive skills involved in learning to read characters in Chinese [1,2]. Most of these studies focused on children in Hong Kong who speak Cantonese and use traditional Chinese script [3]. To date, there is an increasing amount of research that has explored the processes of learning to read in children in Mainland China. Unlike their peers in Hong Kong, these children speak Mandarin and receive formal literacy education starting in Grade 1, while using Pinyin as phoneme-based script in parallel with character instruction in order to facilitate their insight into the correspondences between oral and written language. Pinyin training has been shown to affect children’s phonological awareness [4,5,6], which is found to be related to early Chinese character reading [7,8]. It has also been found that visual skills affect reading ability for children in Hong Kong, but only in kindergarten and not in later phases [9], presumably, as a consequence of growing reading experience [10]. However, results on precursors of learning to read Chinese characters in Mainland China are limited in scope [11,12] and mostly based on concurrent designs [12,13]. How phonological awareness, Pinyin letter knowledge, and visual skills longitudinally predict subsequent character reading remains unclear. Therefore, we followed children from K3 to first grade in Mainland China to examine the contribution of *n* phonological awareness, Pinyin letter knowledge, and visual skills to character reading in the present study.

### 1.1. Phonological Awareness and Chinese Reading

Phonological awareness has been found to be significant to Chinese reading for both traditional and simplified scripts [1,7,13,14,15,16,17]. It has also been shown that the importance of different psycholinguistic units in explaining variance in reading skills depends on the correspondence between the phonology of the spoken language and the orthography [18,19]. Syllables are the basic units of speech in Chinese. Each syllable can be divided into an onset and a rime [20]. In addition, Mandarin Chinese has four different tones. Although Chinese has a relatively simple syllable structure [11], three different aspects of phonological awareness have been emphasized to be important for learning to read Chinese [1,7,16]: syllable awareness, onset–rime awareness, and lexical tone awareness [7,21].

Syllable awareness has widely been used across studies to represent the core of phonological awareness. In cross-sectional studies, it was shown to be strongly associated with Chinese reading in both kindergartners and primary school children in Hong Kong [16,22] and in Mainland China [7,11]. Several studies also found that syllable awareness plays a predictive role from kindergarten to intermediate and higher grades in primary school [23,24,25]. There is only one longitudinal study from Lin et al. [26] that focused on Mainland kindergartners’ Chinese reading development while considering the role of syllable awareness. Results showed that syllable awareness predicted subsequent reading. However, this study did not consider the predictive role of visual skills.

Since Chinese script has a morphosyllabic base, the role of onset–rime awareness playing in Chinese character reading can be called debatable. Accordingly, the evidence that onset–rime awareness is relevant to Chinese character reading can be called inconclusive for Hong Kong children [3,16,27,28], as well as for Mainland children [7,11,13]. 

Finally, awareness of lexical tone is generally considered to be important in Chinese reading [20,29,30], because of the fact that lexical tone may distinguish across the different meanings of a syllable [7]. Tone awareness has been found to be uniquely associated with Chinese character recognition not only for kindergarten children in Hong Kong [3] but also for Mainland Chinese children at the same age [7]. Yin et al. [6] proved the critical role of tone awareness in Chinese reading for children aged 8–9 years old in Mainland China after controlling for rapid naming. However, to our knowledge, there is no longitudinal study that has considered the contribution of tone awareness in Chinese reading. 

### 1.2. Pinyin Skills and Chinese Reading

Although grapheme–phoneme correspondence rules cannot be used to help to read Chinese characters, children in Mainland China receive Pinyin instruction to connect oral Chinese and Chinese print [31]. Pinyin helps reliably and directly with character pronunciation. Various tasks testing Pinyin skills (Pinyin letter name knowledge, invented Pinyin spelling, and Pinyin sentence reading) have shown a positive association with Mandarin Chinese reading [26,31,32,33]. Pinyin letter knowledge has been found to be associated with Chinese word reading in kindergarten [33]. It was also found that invented Pinyin spelling in kindergarten predicted first-grade character reading in Mainland China [26] as well as fourth-grade character reading [34]. Finally, Li et al. [32] showed that Pinyin sentence reading directly impacted first-grade character reading in Mainland China. 

### 1.3. Visual Skills and Chinese Reading

Another factor that is of particular importance for learning to read Chinese is visual skills [33,35]. Although skilled Chinese readers process Chinese characters as a whole rather than separate units [36], beginner’s Chinese character learning typically focuses on copying visual forms [33] and beginner readers in Mainland China are taught to decompose a character into its smaller meaningful stroke patterns [37]. There are six basic strokes in Chinese orthography: a dot, a horizontal line, a vertical line, a diagonal line falling from right to left, a diagonal line falling from left to right, and a stroke with a change in direction [12]. Different numbers of strokes are arranged to make different radicals, which are larger components of Chinese characters. 

Visual skills have been proven to be associated with Chinese reading ability, significantly in the beginning phase of reading acquisition for Hong Kong children [9,38]. 

However, children in Mainland China use the simplified script with Pinyin and receive formal character reading instruction later on, in Grade 1. The difference in context between children in Hong Kong and in Mainland China may lead to different associations between visual skills and Chinese character reading. There do exist studies examining the contribution of visual skills to character reading in Mainland children, but most of them were conducted in concurrent designs [11,12,13] and the results were mixed. Li et al. [11] found that visual spatial relationships and visual memory were not uniquely associated with character reading for children both in kindergarten and primary school, whereas Luo et al. [12] found that visual discrimination predicts Chinese character reading only for kindergarten children but not for children in primary school. The reason could be the different predictors included in these two studies. Luo et al. [12] compared the relative contributions of geometric-figure processing and character-configuration processing to Chinese reading. Therefore, the contribution of geometric-figure processing disappeared when the character-configuration processing was considered, as the latter one required the ability to discriminate the visual features of the components inside Chinese characters. Few studies have considered the longitudinal predictive role of visual skills. McBride-Chang et al. [10] followed kindergarten children for 9 months, both in Hong Kong and in Mainland China. Visual–spatial relationships were shown to be more associated with Chinese character recognition compared with the other two visual skills. Tong et al. [38] followed 4-year-old kindergarteners in Hong Kong for 2 years. Their study showed that the task of visual–spatial relationships predicted word reading in Chinese both concurrently and subsequently. Hulme et al. [15] followed children from Grade 1 to Grade 3 in Mainland China, and showed that visual discrimination predicted initial word reading but not subsequent reading. However, to our knowledge, there is no study that has followed children from kindergarten to first grade to consider the role of visual skills in Chinese reading.

### 1.4. Present Study

Many studies on Chinese character reading have focused on Hong Kong children, who speak Cantonese and use traditional Chinese script. Compared to Mandarin Chinese, Cantonese is more complicated in phonological structure. The numbers of the initial and final consonants are different in the two dialects, and Cantonese has nine tones while Mandarin Chinese only has four tones. The traditional script used in Hong Kong is also more complicated than the simplified Chinese script used in Mainland China. In Mainland China, children receive Pinyin instruction as a bridge to character reading. In the literature review above, phonological awareness, Pinyin knowledge, and visual skills have been found to be associated with character recognition of simplified Chinese script. Although one previous study of Lin et al. [26] from kindergarten to first grade considered the predictive role of syllable awareness and Pinyin letter knowledge, most of the previous research followed concurrent designs. The present study extends this previous work by including two more types of Chinese phonological awareness, which have been proven to be important to Chinese reading: onset–rime awareness and tone awareness; and two types of visual perception skills: visual discrimination skill and visual–spatial relationships skill. 

The first main purpose of the current study was to track the development of children’s character reading and reading-related skills from kindergarten to first grade. To begin with, we examined how children’s phonological awareness, Pinyin letter knowledge, and Chinese character reading developed from kindergarten to first grade. 

Second, we investigated to what extent the variance in children’s Chinese character reading could be explained by their phonological awareness, Pinyin letter knowledge, and visual skills. Specifically, we examined to what extent phonological awareness, Pinyin letter knowledge, and visual skills at kindergarten would longitudinally predict subsequent character reading in first grade, taking into account the effect of its autoregressor. It was hypothesized that kindergarten phonological awareness, Pinyin letter knowledge, and visual skills would each contribute to Chinese reading one year later. 

## 2. Method

### 2.1. Participants

A total of 55 children (27 boys, 28 girls, SD = 2.82) participated in this 1-year longitudinal study. Participants were initially tested in December, which was in the middle of the academic year. Their mean age was 68 months, and they were in Kindergarten year 3. All the participants were tested again 12 months later in first grade. All of the children speak Mandarin and were recruited from a kindergarten associated with a university in central China. One year later, they went to the elementary school associated with the same university. Both the kindergarten and the elementary school were located inside the university, which was situated in an urban area in Mainland China. The parents of the participants were mostly teachers at the university, thus having a high education level.

Kindergarten in Mainland China consists of three years and is independent of primary school. Children do not receive formal literacy instruction until primary school. In kindergarten, children learn basic knowledge from different kinds of activities, such as picture book reading, music activities, outdoor activities, and so forth. Although children do not learn how to read and write in kindergarten, they are exposed to Chinese characters through home literacy activities, digital learning apps, and daily life. At the very beginning of primary school, children focus on how to read and write Pinyin letters. After that, they begin to learn short texts and learn how to read and write Chinese characters. 

### 2.2. Measurements 

#### 2.2.1. Phonological Awareness in Chinese

The syllable deletion task from Shu et al. [7], which consisted of 16 two-syllable words, was used to test syllable awareness. Eight items were real words and the words in the other eight items were nonsense. During the testing process, experimenters pronounced a word and children were then asked to delete the first or the last syllable. The maximum score was 16. The Cronbach’s alpha when assessed in Kindergarten was 0.95. However, it fell to 0.53 when children were in first grade due to a ceiling effect. Therefore, this measure in Grade 1 was not included in the analyses.

Onset- and rhyme-detection tasks, which were adapted from the onset- and rime-detection tasks from the study of Shu et al. [7], were used to test onset and rhyme awareness. There were 20 items in this task. In each item, children were first shown with a picture of the target single-syllable word, and the experimenter made sure that children gave the correct word corresponding with the picture, then children were asked to choose one that had the same onset or rhyme with the target word from four words with corresponding pictures. The Cronbach’s alpha of the task was 0.42 in kindergarten and 0.77 in Grade 1. To increase the reliability of this task at the first testing time, seven items were deleted at that time point, which improved reliability to 0.63.

The tone-discrimination task, which was revised from the tone-detection task of Shu et al. [7], was used to measure children’s tone awareness. At the first measurement, the experimenter orally presented ten pairs of one-syllable characters to the children and asked if the characters in each pair had the same or different tones. To effectively measure children’s tone sensitivity, six pairs of characters had different Pinyin spellings (same rimes but different onsets) and different tones, such as 刀 [dāo] (meaning: knife) and 帽 [mào] (meaning: hat), while the other four pairs of characters had different Pinyin spellings (same rimes but different onsets) but the same tones, such as 椅 [yǐ] (meaning: chair) and 笔 [bǐ] (meaning: pen). The Cronbach’s alpha of this task was low: 0.31. At the second measurement, in Grade 1, ten items were added to this task. Five pairs of characters were exactly the same in their Pinyin spellings and tones, such as 灰 [huī] and灰 [huī] (meaning: gray), while the other five pairs of characters had the same Pinyin spellings but different tones, such as 油 [yóu] (meaning: oil) and 友 [yǒu] (meaning: friend). The Cronbach’s alpha of this task then improved to 0.77.

#### 2.2.2. Pinyin Letter Knowledge

Pinyin letter knowledge was tested via a Pinyin naming task. Children were presented with a card, on which 47 Pinyin letters were randomly ordered in rows, and were asked to read the letters aloud. At the first measurement, most of the participants (61.8%) did not know any letters, and thus scored zero in this task. The average mean was low: 2.76. We therefore transformed the scores into a dummy variable, with 1 representing those children who scored above the mean, and 0 to represent those children who had lower scores than the mean. At the second measurement point, in Grade 1, all of the children had a maximum score of 47 in this task. Because of the ceiling effect at the second point, this measure was not included in the regression analysis.

#### 2.2.3. Visual Perception Skills

Two tasks, namely the visual discrimination task and visual–spatial relationship task, were used to test children’s visual perception skills. There were one practice item and sixteen test items in each task.

Visual discrimination was measured using one subtest from Gardners’s Test of Visual-Perceptual Skills Revised [39] at the first measurement. In this task, children were shown a target form (e.g., 
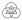
) alongside alternative five visually similar choices (e.g., 
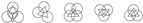
). They were asked to choose the form that was identical to the target one. Testing was terminated if children failed on five consecutive items. The maximum score of this task was 16, and the Cronbach’s alpha was 0.80, which was excellent.

The test of children’s visual–spatial relationships was measured by a subtest from Gardner’s Test of Visual-Perceptual Skills Revised [39]. For each item, experimenters presented five forms (e.g., 
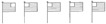
), of which one target form was different in direction from the other four forms. Children were asked to detect the target one. Testing stopped if children failed on five consecutive items. The maximum score was 16, and Cronbach’s alpha was 0.88, which was considered excellent.

#### 2.2.4. Chinese Character Reading

Chinese character recognition task [7], which consisted of 60 single characters, was used to measure Chinese character reading. All the characters were selected from textbooks in first grade and were listed in an increasing level of difficulty. The test stopped after 10 consecutive mistakes. The Cronbach’s alpha of this task at the two measurements (kindergarten, Grade 1) were 0.98 and 0.91, respectively.

### 2.3. Procedure

Consent letters from the parents and the school were obtained, and convenient time for testing the children was arranged individually. Children were tested on the measures in both kindergarten and one year later at primary school by trained preschool education major students. All the experimenters were native Mandarin Chinese speakers. Assessment in kindergarten took about 50 min. In Grade 1, all tests were again conducted, with the exception of the visual perception tasks, because previous research has shown that this ability does not increase from kindergarten to first grade [12]. The assessment this time took about 35–40 min.

## 3. Results

### 3.1. Descriptive Statistics

Means, standard deviations, ranges, skewness, and kurtosis for all the measurements are displayed separately for kindergarten and Grade 1 level in Table 1. Tone awareness in Grade 1 was not normally distributed. As recommended by Field [40], log transformation was used to transform this variable in kindergarten and Grade 1 to normal distribution. The average scores of syllable awareness and Pinyin letter knowledge tasks were at the ceiling in Grade 1.

We first examined whether children improved from kindergarten to Grade 1. Paired samples t-tests revealed a significant improvement in onset–rime awareness—*t* (54) = 9.32, *p* < 0.001, Cohen’s d = 1.26; tone awareness—*t* (54) = 7.33, *p* < 0.001, Cohen’s d = 0.99; and Chinese character recognition—*t* (54) = 10.48, *p* < 0.001, Cohen’s d = 1.41. Since syllable awareness and Pinyin letter knowledge were not normally distributed, nonparametric Wilcoxon signed-rank tests [41] were carried out to compare these variables in the two different measurements. Results also showed that scores improved across these two tasks: syllable awareness—*Z* = −3.75, *p* < 0.001; and Pinyin letter knowledge—*Z* = −6.65, *p* < 0.001.

Table 2 presents the correlations between all the measurements. Syllable and tone awareness positively correlated with kindergarten character reading. Both of the two visual skills tasks were significantly associated with Chinese character reading in kindergarten. There were no significant correlations between the other variables and Chinese character reading in kindergarten.

With regard to the relationships between kindergarten variables and Grade 1 Chinese character reading, none of the phonological awareness measures in kindergarten were associated with Grade 1 Chinese character reading. Moreover, kindergarten Pinyin letter knowledge did not correlate with Grade 1 Chinese character reading. However, kindergarten visual discrimination significantly correlated with Grade 1 Chinese character reading, and visual–spatial relationships also correlated with Grade 1 Chinese character reading. Kindergarten Chinese character recognition had a significant correlation with Grade 1 character recognition.

### 3.2. Predicting Grade 1 Chinese Character Reading

Next, we performed a regression analysis to examine the question of to what extent kindergarten variables would predict Grade 1 Chinese character reading. Chinese character reading in first grade was included as the dependent variable. Phonological awareness, Pinyin letter knowledge, and the two visual perception skills in kindergarten were entered as the independent variables. The model was significant, explaining 16% of the variance in Grade 1 Chinese character reading, *F* (6, 48) = 2.65, *p* = 0.03 (see Table 3). Of interest, across all the variables, only the final beta weight for visual discrimination was significant, which means visual discrimination uniquely predicted Grade 1 character reading, while phonological awareness and Pinyin letter knowledge did not.

We next checked whether the effect would hold if we control for the autoregressor. We thus performed a hierarchical regression analysis. The character recognition in kindergarten was entered in the first step, and the other variables were included in the second step. Table 3 shows the results: Kindergarten character reading predicted 62% of total variance in the character reading in first grade—*F* (1, 53) = 89.63, *p* < 0.01. The predictors in the second step were not uniquely associated with Chinese character reading in Grade 1—ΔR² = 0.04, *F* (6, 47) = 0.91, *p* = 0.50. In all, the results may suggest a mediating role of kindergarten character reading in the relation between visual discrimination skill and Grade 1 character reading.

### 3.3. The Mediating Role of Kindergarten Chinese Character Reading

To examine this, we used the Process add-on in SPSS [42] to conduct a mediation analysis. Kindergarten visual discrimination was entered as the independent variable, Grade 1 character reading was included as the dependent variable, and kindergarten character reading was entered as the mediator. Bootstrapping was set at 5000 cycles.

The total R^2^ of the model was 0.16 (*p* = 0.003), which means visual discrimination indirectly affects Grade 1 character reading via its effect on kindergarten character reading (ab = 0.41, CI = [0.12, 0.70]). The total effect from kindergarten visual discrimination to character reading in first grade was significant (c = 0.69, *t* (53) = 3.13, *p* < 0.01), but the direct effect was not significant (c’ = 0.28, *t* (52) = 1.85, *p* = 0.07). Results are depicted in Figure 1, reporting unstandardized coefficients.

## 4. Discussion

The present study explored the predictors of early Chinese character recognition in 55 kindergarten children in Mainland China. In the first step, we investigated the development of phonological awareness, Pinyin letter knowledge, and character reading from the last year of kindergarten to first grade. In addition, we examined to what extent children’s character reading in first grade could be predicted from their phonological awareness, Pinyin letter knowledge, and visual perception skills as measured in kindergarten. Our study is possibly the first longitudinal study to examine how phonological awareness, Pinyin letter knowledge, and visual perception skills in kindergarten predict subsequent character reading for Chinese children in Mainland China. We found that children improved significantly in phonological awareness, Pinyin letter knowledge, and Chinese character recognition from kindergarten to first grade. Kindergarten visual discrimination was the unique predictor of character recognition in Grade 1, but this effect disappeared if kindergarten character recognition was taken into account as an autoregressor. Moreover, it was evidenced that visual discrimination had a significant indirect effect on Grade 1 character recognition through character recognition in kindergarten.

Consistent with our hypothesis, children’s phonological awareness, Pinyin letter knowledge, and character recognition improved significantly from the last year of kindergarten to Grade 1. The low scores in Pinyin letter knowledge in kindergarten are in line with a previous study by Wang et al. [33]. A striking finding was that the children reached a ceiling effect at the beginning of first grade, suggesting mastering of this knowledge at a rapid speed. One apparent reason for this is that Mainland Chinese children are being taught Pinyin letter knowledge intensively in the first two months of Grade 1. As expected, children’s phonological awareness and character recognition also improved over time. This corresponds with a study of Shu et al. [7], testing phonological awareness level in four grades: K1–K3 and first grade. They showed that phonological awareness increased gradually and steadily. Pinyin practice could be another factor to support the development of phonological awareness and character recognition. In the present study, all of the participants had the maximum score in Pinyin letter knowledge in first grade, so we could not examine the relationship between Pinyin letter knowledge, phonological awareness, and character reading for Grade 1 children. However, it has been proven that Pinyin skills strengthen the development of phonological awareness [7,16] and are related to reading [9,13,33].

With respect to the association between the kindergarten phonological awareness and early character reading in first grade, our study showed that first-graders’ character recognition could not be predicted from kindergarten phonological awareness. This finding suggested that phonological awareness was not relevant to Mandarin Chinese character reading, at least for early primary schoolers. The study of Li et al. [11] showed that rime detection was not associated with Chinese reading for kindergarteners, but it was moderately associated with this measurement for primary school children. Onset and rime are represented in the phonetics across characters, and this linkage may become increasingly clear to older Chinese readers [7]. Our results provided further evidence, combined with previous studies, that phonological awareness may play different roles in Chinese character reading at different learning stages. It is relevant in kindergarten and in more advanced readers, but not for children who are just learning how to read. Shu et al. [7] found that for a group of children from K1 to K3 in Mainland China, Chinese character recognition was explained by syllable and tone awareness. Onset–rime awareness was shown not to be associated with reading in Mainland Chinese students in Grade 1 but in Grade 2 and 5 [13]. The study of Pan et al. [25] also found that preliterate syllable awareness significantly predicted reading at age 11 years. It can tentatively be concluded that children may rely more on phonological awareness to recognize characters in the preliterate stage and higher grades in primary school, but not at the beginning of learning to read Chinese. The reason could be that for younger children who are preliterate, Pinyin knowledge can help them to be able to make the connection between speech and the print, whereas older children in higher grades in primary school may rely more on phonetic radicals to provide phonological cues to read Chinese with the development of their reading experience [43].

The present study further showed that Pinyin letter knowledge in kindergarten did not predict Chinese character reading one year later. There do exist studies that proved the important role of Pinyin skills for Chinese reading [13,26,31,33]. In the study of Lin et al. [26], K-3 children were followed for one year, just as what we did in the present study. However, their study considered the predictive role of both invented Pinyin spelling and Pinyin letter knowledge. The result was that invented Pinyin spelling predicted Chinese word reading longitudinally, whereas Pinyin letter knowledge did not. It is important to note that the Pinyin skill task that was used in most studies was to ask children to read the Pinyin of 60 meaningful syllables or write down words in Pinyin, but not just to recognize single Pinyin letters as we did in the current study.

With regard to the influence of preliterate visual skills on the development of character reading, we showed the important role of kindergarten visual discrimination in subsequent character reading. Visual discrimination skill reflects children’s ability to match or determine exact characteristics of two forms when one of the forms is among similar forms [39]. This skill is crucial for early Chinese character reading, because children in first grade usually start to learn from simple characters, which are sometimes visually similar, such as 大 (meaning “big”) and 太 (meaning “too”) and they have to rely on visual discrimination to distinguish the tiny differences between characters. This result may also support the idea that the logographic stage, in which there is a link between visual skills and character reading, appears to emerge later for Mainland Chinese children compared to Hong Kong children. Children in Hong Kong receive formal literacy instruction from kindergarten, while children in Mainland China receive it from first grade. Lin et al. [9] found that for Hong Kong kindergarten children, there exists a bidirectional relationship between visual spatial skill and Chinese character reading from K1 to K2, but a nonsignificant association for such relationship from K2 to K3. Therefore, the stage at which visual skills affect character reading appears later for Mainland Chinese children compared with Hong Kong children. We also found that kindergarten character recognition has a significant direct effect on children’s character recognition in first grade. The difficult orthography of Mandarin Chinese might explain this result. Due to the deep nature of Chinese orthography, literacy education is sufficient in the acquisition of Chinese literacy. Therefore, children who begin to learn to read at an early age perform better in reading than those who learn to read later. It is important to note that visual discrimination influences Chinese character recognition in Grade 1 mainly through its contribution to character recognition in kindergarten.

There were several limitations in this study. First, we only used one task to test Pinyin skills, and unfortunately, the measure was easy for first-grade children and reached a ceiling effect. It would be interesting to include different tasks of Pinyin skills and follow the development of Pinyin skills. Second, visual skills were only tested in kindergarten, as it was regarded as an ability that would not be influenced through teaching in first grade. Testing this measure in both kindergarten and first grade will help us to know the concurrent relationship between visual skills and character reading for first grade. Third, the sample size was relatively small, restricting us from exploring the deep relationship between all the variables, and the homogeneity of the family background of children may limit the generalization of the results. In future research, more children from diverse types of families need to be included.

In summary, the findings in the present study expanded our understanding of the relationships between phonological awareness, Pinyin letter knowledge, visual perception skills, and Chinese character reading. By longitudinally tracking children in Mainland China from the last year of kindergarten to first grade, the present study showed that these children develop in phonological awareness, Pinyin letter knowledge, and Chinese character reading. The present study also underscores the importance of visual discrimination for Chinese character reading development.

### Implications

The findings of the present study first suggest the relevance of visual skills in early reading development. This can be used as a means to improve reading education. This can be done by paying specific attention to the discriminating features of the different characters, for example, when children are asked to copy a character in writing classes. Teachers may pay explicit attention to character formation, character size, slant control, and orientation. The more precise children have the internal mental representations of the composition of the characters, the easier children recognize them. In studies focusing on alphabetic languages, it was found that supporting writing within the kindergarten curriculum improved chidren’s visual skills and emergent literacy skills [44,45]. In this view, early basic Chinese writing skill instruction can also help children to effectively discriminate subcharacter components, thus recognizing whole characters. Of course, the benefits of such educational instruction for young children in Mainland China need to be confirmed by further research.

The finding that kindergarten character reading uniquely predicted character reading in first grade implies that introducing early literacy education may facilitate Chinese character reading development for children in Mainland China. Considering the fact that formal literacy teaching is not allowed in kindergartens in Mainland China, informal literacy practice becomes more valuable. Enriching the literacy learning environment, for example, by increasing teacher–children interaction in classroom literacy activities, such as shared book reading [46], and learning the composition of Chinese characters in games [47,48], could be effective educational practices.

## Figures and Tables

**Figure 1 children-09-01946-f001:**
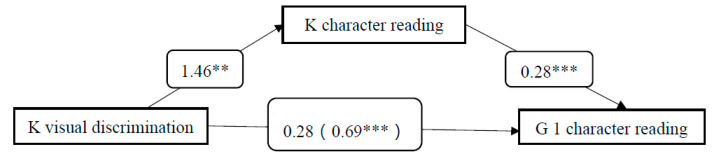
Mediation model of kindergarten visual discrimination, kindergarten character reading, and Grade 1 character reading. Note. The total effect © is between brackets, outside the brackets are the direct effects (c’). K represents kindergarten and G 1 represents Grade 1. ** *p* < 0.01, *** *p* < 0.001.

**Table 1 children-09-01946-t001:** Means, standard deviations, range, skewness and kurtosis of phonological awareness, Pinyin letter knowledge, visual perception skills, and Chinese character reading of 55 children in kindergarten and Grade 1 (*n* = 55).

	Kindergarten					Grade 1						
	M	SD	Range	Skewness	Kurtosis	M	SD	Range	Skewness	Kurtosis	*t*	*Z*
Syllable awareness	14.62	3.36	0–16	−3.45	12.39	15.96	0.27	14–16	−7.42	55		−3.75
Onset–rime awareness	0.48	0.21	1–13	0.32	−0.22	0.77	0.18	6–20	−0.77	0.12	9.32 ***	
Tone awareness	0.63	0.18	1–10	−0.21	0.68	0.86	0.14	4–20	−0.44	0.31	7.33 ***	
Pinyin letter knowledge	2.76	5.69	0–27	2.64	7.09	47	0	47	-	-	-	−6.65
Visual discrimination	9.62	3.69	0–16	−0.61	0.30	-	-	-	-	-	-	
Visual–spatial relationships	11.24	4.19	0–16	−1.19	0.80	-	-	-	-	-	-	
Chinese character reading	33.98	16.97	5–59	−0.28	−1.21	51.62	6.46	33–60	−0.65	−0.25	10.48 ***	

Note. The proportion of onset–rime and tone awareness tasks was used in the table. *** *p* < 0.001.

**Table 2 children-09-01946-t002:** Correlations for phonological awareness, Pinyin letter knowledge, visual perception skills, and Chinese character reading in kindergarten and first grade.

	1	2	3	4	5	6	7	8	9	10
1. K syllable awareness	-									
2. K onset–rime awareness	0.08	-								
3. K tone dummy	0.04	0.23	-							
4. K Pinyin dummy	0.10	0.15	0.37 **	-						
5. K visual discrimination	0.35 **	0.42 **	0.16	−0.02	-					
6. K visual–spatial relationships	0.25	0.43 **	0.32 *	0.14	0.65 **	-				
7. G 1 onset–rime awareness	0.19	0.27 *	0.17	−0.02	0.47 **	0.58 **	-			
8. G 1 tone awareness	0.20	0.03	−0.07	−0.02	0.20	0.18	0.39 **	-		
9. K character reading	0.36 **	0.05	0.33 *	0.24	0.32 *	0.34 *	0.21	0.13	-	
10. G 1 character reading	0.26	0.05	0.26	0.16	0.40 **	0.29 *	0.27 *	0.12	0.79 **	-

Note. K represents kindergarten and G 1 represents Grade 1. K tone dummy variable and K Pinyin dummy variable were included in Table 2. * *p* < 0.05; ** *p* < 0.01.

**Table 3 children-09-01946-t003:** Hierarchical regression explaining Grade 1 Chinese character reading from K phonological awareness, Pinyin letter knowledge, and visual perception skills.

		Model without K Character Reading in the First Step				Model with K Character Reading in the First Step			
Step	Predictor	Δ R²	B	SE	ß	Δ R²	B	SE	ß
1.	K character reading		-	-	-	0.63 ***	0.30	0.03	0.77 ***
2.	K syllable awareness	0.25 *	0.21	0.26	0.11				
	K onset–rime awareness		−0.46	0.34	−0.20				
	K tone awareness		2.59	1.97	0.19				
	K Pinyin letter knowledge		1.79	2.08	0.12				
	K visual discrimination		0.72	0.31	0.41 *				
	K visual–spatial relationships		0.01	0.27	0.01				
Total	R² adj	0.16 *				0.62 ***			

Note. K Pinyin dummy variable was included in Table 3. * *p* < 0.05; *** *p* < 0.001.

## Data Availability

The data presented in this study are available on request from the corresponding author.

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
