# Peer review of "Predictors of Early Mandarin Chinese Character Reading Development"

_children, 2022, doi:10.3390/children9121946_

Round 1
Reviewer 1 Report
Dear Authors,
I greatly appreciate having the opportunity to review the paper titled “Predictors of Early Mandarin Chinese Character Reading Development”. This study’s focus on the development of Mandarin Chinese character reading 9 development and its predictors in 55 children from kindergarten to first grade in Mainland China is very interesting. The report is logically organized, comprehensive, and well-written. The authors are uniquely positioned to conduct this study. However, even if am enthusiastic about the focus of this paper, I felt that how the work is presented merits some clarification and further elaboration and this more specifically for example regarding the lack of more in-depth info on students’ and school characteristics, frameworks used, data collection procedure and analysis and the need for more nuance when it comes to the discussion and conclusions. What with the added value of the insights based on this paper, the key message? How will this more specifically improve character 19 reading ability (education)?
Please revise the abstract, and make it more condensed and coherent. Please also give methodological information (design, N, …).
Introduction:
-Please compose your text better from the very start. Also pay more attention to the coherence between sentences, paragraphs, and so on.
-Please reread the research questions carefully. What info do you give in the introduction on effective ways to promote reading and how do you relate this to your data collection procedure, data analysis, and so on?
Method:
-How representative are your subjects?
-Validity, reliability and objectivity of this study?
-What with the chosen schools/students?
-What with the questions asked? Link with frameworks? Protocol?
- Could you please give more insight into the data analysis (path analysis procedure)?
Results:
-Can it be that the small number of schools/students and the type of schools and community where they were situated are not representative enough?
- -It feels sometimes that the results given are the result of some cherry picking.
Discussion:
-Is this really a discussion? It sometimes reads more as a continuation of the results.-Elaborate more on the findings, relating them to the frameworks in the introduction, and the research questions. Bring more overall coherence in your text.
Limitations:
-Please elaborate more on the limitations, because they are really there.
What are the conclusion(s)?
Author Response
I greatly appreciate having the opportunity to review the paper titled “Predictors of Early Mandarin Chinese Character Reading Development”. This study’s focus on the development of Mandarin Chinese character reading 9 development and its predictors in 55 children from kindergarten to first grade in Mainland China is very interesting. The report is logically organized, comprehensive, and well-written. The authors are uniquely positioned to conduct this study. However, even if am enthusiastic about the focus of this paper, I felt that how the work is presented merits some clarification and further elaboration and this more specifically for example regarding the lack of more in-depth info on students’ and school characteristics, frameworks used, data collection procedure and analysis and the need for more nuance when it comes to the discussion and conclusions. What with the added value of the insights based on this paper, the key message? How will this more specifically improve character 19 reading ability (education)?
Response: Thank you very much for the positive review.
Please revise the abstract, and make it more condensed and coherent. Please also give methodological information (design, N, …).
Response: We have revised this part. Now we focused on the two research questions. One is the development of PA, Pinyin letter knowledge and Chinese character reading from kindergarten to Grade 1; and the other one is to what extend children’s Chinese character in first grade could be predicted from the kindergarten variables.
Introduction:
-Please compose your text better from the very start. Also pay more attention to the coherence between sentences, paragraphs, and so on.
-Please reread the research questions carefully. What info do you give in the introduction on effective ways to promote reading and how do you relate this to your data collection procedure, data analysis, and so on?
Response: We have revised this part and added more information. Please see the revised manuscript.
Method:
-How representative are your subjects?
-Validity, reliability and objectivity of this study?
-What with the chosen schools/students?
-What with the questions asked? Link with frameworks? Protocol?
- Could you please give more insight into the data analysis (path analysis procedure)?
Response: This is one limitation in the present study, we have added it in the Discussion.
“Third, the sample size was relatively small, restricting us to explore the deep relationship between all the variables. And the homogeneity of the family background of children may limit the generalization of the results. n future research, more children from diverse types of families need to be included.”
Because this study was part of a cross-cultural project, which was conducted in both China and the Netherlands, collecting data in two different countries and the longitudinal designs brought us lots of challenges. We only collected data in the kindergarten and the primary school associated with the university in central China.
The relatively small sample size is indeed a limitation in the present study and restricted us to use path analysis, that is acknowledged in the Discussion. As kindergarten and primary school in China are separated educational institutions, which makes it challenging to follow children for one year. It is needed to included more participants in the future study.
Results:
-Can it be that the small number of schools/students and the type of schools and community where they were situated are not representative enough?
- -It feels sometimes that the results given are the result of some cherry picking.
Response: We have included this limitation in the Discussion. In future research, more children from diverse types of families need to be included.
Discussion:
-Is this really a discussion? It sometimes reads more as a continuation of the results. Elaborate more on the findings, relating them to the frameworks in the introduction, and the research questions. Bring more overall coherence in your text.
Response: We have followed your comments and revised the manuscriopt.
Limitations:
-Please elaborate more on the limitations, because they are really there.
What are the conclusion(s)?
Response: We have followed your comments and elaborated more on the limitations, please see the revised manuscript.
Reviewer 2 Report
Judging from the procedure of argumentation and the research results, the conclusion of this paper is credible. The shortcomings of the research, the author has pointed out at the end of the article.
Author Response
Thank you very much for the positive comments!
Reviewer 3 Report
This is an interesting longitudinal study on roles of neuro-cognitive and pre-literacy skills in predicting early Mandarin Chinese character reading in Chinese kindergarten to first graders. Strengths of the study includes the two-wave longitudinal design, use of objective measures to assess key constructs, and the paper is nicely written.
Introduction
1. The authors reviewed that much prior research on links of phonological awareness to Chinese reading was conducted with Cantonese speaking children (e.g., children in Hong Kong). But they did not explain the key differences in reading development in Mandarin and Cantonese readers, which might include both linguistic differences between Mandarin and Cantonese, differences between simplified and traditional Chinese characters, and differences in reading curriculums between Hong Kong and Mainland Chinese kindergarten and elementary schools. It would be useful to summarize the similarities and differences in reading development between the two subpopulations (Hong Kong children and Mainland Chinese children).
2. On p. 3, line 102 to 116, the authors reviewed a few existing studies on the concurrent links between visual skills and Mandarin character reading. Instead of summarizing each study results in sequence, it would be more useful to synthesize the study results and point out the consistences and inconsistence, the potential reasons for the mixed findings, and key gaps in this literature. This will help readers see the new contributions of the proposed study and how it advances the existing literature.
Methods
3. It is important to point out that the sample consisted of children from highly educated families (university kindergarten and elementary school) and discuss to what degree the findings from the present study generalize to Mandarin readers from other socioeconomic backgrounds.
4. It is helpful that the authors provided some background information on the literacy instruction in kindergarten in Mainland China. Although children do not receive formal literacy instruction in kindergarten, children might receive literacy instruction outside school (e.g., parents, enrichment classes, weekend reading lessons...). I wonder whether there is any data on home literacy environment and how variations in home literacy environment can impact the findings;
5. There are some measurement issues in the study instruments that need to be addressed in preliminary analyses. Measures that showed poor reliabilities or ceiling effects in the sample should be dropped from analyses.
1) The measure of Tone Awareness at T1 had poor reliability (0.31). This variable should be dropped from subsequent analyses;
2) Several measures reached ceiling (no variances) in T2: Pinyin letter knowledge and the visual skill tests. These variables should be dropped from subsequent analyses:
3) Syllable awareness was negatively skewed at both time points: it should be transformed or analyzed using nonparametric methods.
Given the ceiling effects in many of the Grade 1 measures, I am not sure the study aim 1 (testing concurrent relations between Grade 1 phonological awareness, Pinyin letter knowledge, and visual skills and Grade 1 character reading) can be tested with robust method. The authors might consider dropping this aim and focus on the aims testing longitudinal relations (kindergarten variables predicting Grade 1 reading).
Author Response
This is an interesting longitudinal study on roles of neuro-cognitive and pre-literacy skills in predicting early Mandarin Chinese character reading in Chinese kindergarten to first graders. Strengths of the study includes the two-wave longitudinal design, use of objective measures to assess key constructs, and the paper is nicely written.
Response: Thank you very much for the positive review.
Introduction
- The authors reviewed that much prior research on links of phonological awareness to Chinese reading was conducted with Cantonese speaking children (e.g., children in Hong Kong). But they did not explain the key differences in reading development in Mandarin and Cantonese readers, which might include both linguistic differences between Mandarin and Cantonese, differences between simplified and traditional Chinese characters, and differences in reading curriculums between Hong Kong and Mainland Chinese kindergarten and elementary schools. It would be useful to summarize the similarities and differences in reading development between the two subpopulations (Hong Kong children and Mainland Chinese children).
Response: We have followed your comments and added this information in the Present Study.
- On p. 3, line 102 to 116, the authors reviewed a few existing studies on the concurrent links between visual skills and Mandarin character reading. Instead of summarizing each study results in sequence, it would be more useful to synthesize the study results and point out the consistence and inconsistence, the potential reasons for the mixed findings, and key gaps in this literature. This will help readers see the new contributions of the proposed study and how it advances the existing literature.
Response: We have followed your comments and revised them. Please see the revised version.
Methods
- It is important to point out that the sample consisted of children from highly educated families (university kindergarten and elementary school) and discuss to what degree the findings from the present study generalize to Mandarin readers from other socioeconomic backgrounds.
Response: We have added this as a limitation in the discussion part.
- It is helpful that the authors provided some background information on the literacy instruction in kindergarten in Mainland China. Although children do not receive formal literacy instruction in kindergarten, children might receive literacy instruction outside school (e.g., parents, enrichment classes, weekend reading lessons...). I wonder whether there is any data on home literacy environment and how variations in home literacy environment can impact the findings;
Response: We have followed your comments and added the relevant information. We did not test home literacy environment. It would be very interesting to include this information in the future study.
- There are some measurement issues in the study instruments that need to be addressed in preliminary analyses. Measures that showed poor reliabilities or ceiling effects in the sample should be dropped from analyses.
1) The measure of Tone Awareness at T1 had poor reliability (0.31). This variable should be dropped from subsequent analyses;
2) Several measures reached ceiling (no variances) in T2: Pinyin letter knowledge and the visual skill tests. These variables should be dropped from subsequent analyses:
3) Syllable awareness was negatively skewed at both time points: it should be transformed or analyzed using nonparametric methods.
Response: Thanks for the suggestions. The tone awareness was included in the regression analysis, because it has been proved to be an important predictor of Chinese character reading, and there is no difference including tone awareness or not in the model. With regard to the ceiling effect of Pinyin letter knowledge and the visual skills in T2, we only used the variables in kindergarten (T1) to predict Chinese character reading in first grade (T2),the two variables were not included in the regression model. The syllable awareness had been transformed in the regression analysis; however, we still used the original scores in Table 1 to let the readers to know how children performed in each task.
Given the ceiling effects in many of the Grade 1 measures, I am not sure the study aim 1 (testing concurrent relations between Grade 1 phonological awareness, Pinyin letter knowledge, and visual skills and Grade 1 character reading) can be tested with robust method. The authors might consider dropping this aim and focus on the aims testing longitudinal relations (kindergarten variables predicting Grade 1 reading).
Response: We have followed your comments and focused on the longitudinal predictive model. Please see the revised version.
Round 2
Reviewer 1 Report
Since the authors have revised the article based on recommendations, I think the paper is ready for publication.
Author Response
-
Thank you for your comments.